# Age and Chronodisruption in Mouse Heart: Effect of the NLRP3 Inflammasome and Melatonin Therapy

**DOI:** 10.3390/ijms23126846

**Published:** 2022-06-20

**Authors:** Marisol Fernández-Ortiz, Ramy K. A. Sayed, Yolanda Román-Montoya, María Ángeles Rol de Lama, José Fernández-Martínez, Yolanda Ramírez-Casas, Javier Florido-Ruiz, Iryna Rusanova, Germaine Escames, Darío Acuña-Castroviejo

**Affiliations:** 1Departamento de Fisiología, Facultad de Medicina, Instituto de Biotecnología, Centro de Investigación Biomédica, Parque Tecnológico de Ciencias de la Salud, Universidad de Granada, 18016 Granada, Spain; sol92@correo.ugr.es (M.F.-O.); ramy.kamal@vet.sohag.edu.eg (R.K.A.S.); josefermar@ugr.es (J.F.-M.); yolandaramirez@correo.ugr.es (Y.R.-C.); jflorido@correo.ugr.es (J.F.-R.); irusanova@ugr.es (I.R.); gescames@ugr.es (G.E.); 2Department of Pediatrics, Division of Hematology-Oncology, Greehey Children’s Cancer Research Institute, University of Texas Health Science Center San Antonio, San Antonio, TX 78229, USA; 3Centro de Investigación Biomédica en Red Fragilidad y Envejecimiento Saludable (CIBERfes), Instituto de Investigación Biosanitaria de Granada (Ibs), 18012 Granada, Spain; 4Department of Anatomy and Embryology, Faculty of Veterinary Medicine, Sohag University, Sohag 82524, Egypt; 5Departamento de Estadística e Investigación Operativa, Facultad de Ciencias, Universidad de Granada, 18071 Granada, Spain; yroman@ugr.es; 6Chronobiology Lab, Department of Physiology, College of Biology, University of Murcia, Mare Nostrum Campus, IUIE, IMIB–Arrixaca, 30100 Murcia, Spain; angerol@um.es; 7UGC de Laboratorios Clínicos, Hospital Universitario San Cecilio, 18016 Granada, Spain

**Keywords:** melatonin, clock genes, chronodisruption, rhythm, aging, inflammaging, NLRP3 inflammasome, mouse heart

## Abstract

Age and age-dependent inflammation are two main risk factors for cardiovascular diseases. Aging can also affect clock gene-related impairments such as chronodisruption and has been linked to a decline in melatonin synthesis and aggravation of the NF-κB/NLRP3 innate immune response known as inflammaging. The molecular drivers of these mechanisms remain unknown. This study investigated the impact of aging and NLRP3 expression on the cardiac circadian system, and the actions of melatonin as a potential therapy to restore daily rhythms by mitigating inflammaging. We analyzed the circadian expression and rhythmicity of clock genes in heart tissue of wild-type and NLRP3-knockout mice at 3, 12, and 24 months of age, with and without melatonin treatment. Our results support that aging, NLRP3 inflammasome, and melatonin affected the cardiac clock genes expression, except for *Rev-erbα*, which was not influenced by genotype. Aging caused small phase changes in *Clock*, loss of rhythmicity in *Per2* and *Rorα*, and mesor dampening of *Clock*, *Bmal1*, and *Per2*. NLRP3 inflammasome influenced the acrophase of *Clock*, *Per2*, and *Rorα*. Melatonin restored the acrophase and the rhythm of clock genes affected by age or NLRP3 activation. The administration of melatonin re-established murine cardiac homeostasis by reversing age-associated chronodisruption. Altogether, these results highlight new findings about the effects aging and NLRP3 inflammasome have on clock genes in cardiac tissue, pointing to continuous melatonin as a promising therapy to placate inflammaging and restore circadian rhythm in heart muscle. Additionally, light microscopy analysis showed age-related morphological impairments in cardiomyocytes, which were less severe in mice lacking NLRP3. Melatonin supplementation preserved the structure of cardiac muscle fibers in all experimental groups.

## 1. Introduction

Heart disease is the leading cause of death worldwide. The main risk factor for cardiovascular diseases (CVDs) is aging [1,2]. The disruption of circadian rhythms and the dysregulation of the immune response are primary hallmarks of aging that manifest cardiac dysfunctions [3].

Circadian rhythms are recurring patterns in a host of behavioral, physiological, and biochemical factors that display periods of near 24 h. The mammalian circadian rhythms are controlled by clock genes [4]. Physiologically, a central pacemaker located in the suprachiasmatic nuclei (SCN) of the hypothalamus regulates the oscillating neuronal and humoral signals in peripheral tissues [5]. Clock genes are essential for controlling cardiac circadian variables such as heart rate, blood pressure and cardiac output [6]. Disruption of the circadian rhythm favors the development of age-related CVDs [7,8,9].

Chronodisruption is defined as a harmful split of a physiological nexus of internal and external times, being a critical link in the cause of chronic disease by a disturbance of the circadian organization of endocrinology, metabolism, physiology, and behavior [10]. Chronodisruption is associated with immune system alterations and inflammatory responses that occur with aging [11]. Inflammaging is a low-level proinflammatory state characterized by subclinical, asymptomatic, chronic, and systemic inflammation. It plays a role in the development and progression of cardiac pathologies and other age-related disorders [12,13]. The main components of innate immunity are NF-kB and NLRP3 inflammasome. NLRP3 inflammasome has been connected to the pathophysiology of several cardiovascular dysfunctions, including heart failure, atherosclerosis, hypertension, myocardial ischemia, cardiomyopathy, and infectious cardiac diseases such as septic myocardial injury, viral myocarditis, and cardiac parasitism [14,15,16,17,18,19,20,21,22]. We have recently elucidated mechanisms by which NLRP3 contributes to cardiovascular disorders, including mitochondria dynamics, mitochondria ultrastructure, intrinsic apoptosis, hypertrophy, and cardiac sarcopenia [23,24].

Melatonin (N-acetyl-5-methoxytryptamine, aMT) is synthesized by the pineal gland [25] as well as most organs and tissues, including the heart [26]. Age-related decline in chronobiotic actions of pineal melatonin has been associated with chronodisruption, inflammaging, and cardiac pathologies [27,28,29,30]. Extrapineal melatonin possesses antioxidative, anti-inflammatory, and mitochondrial homeostatic actions [31]. In experimental models including chronic and acute inflammation, and aging in mouse heart, melatonin diminished the innate immune response, counteracted oxidative stress and improved cardiac mitochondria activity [32,33,34,35]. 

The influence of NF-kB on the disruption of circadian rhythms during aging is well-established [36,37,38], meanwhile, the participation of NLRP3 in the chronodisruption that accompanies aging is unknown. This study aimed to assess the effect of aging and NLRP3 inflammasome on the cardiac circadian system, and to investigate melatonin as a novel therapeutic approach.

## 2. Results

### 2.1. Time Point, Aging, Melatonin Treatment and, to a Lesser Extent NLRP3 Inflammasome, Had Significant Effects on the Expression of Clock Genes in Cardiac Tissue

The expression of the genes *Clock* (Figure 1a), *Bmal1* (Figure 1b), *Per2* (Figure 1c), *Chrono* (Figure 2a), *Rev-erbα* (Figure 2b), and *Rorα* (Figure 2c) was assessed by a multifactorial-ANOVA analysis (Table 1). The test revealed a main effect of time point, age, genotype, and a significant time point × age × genotype interaction for all genes, except for *Rev-erbα*, where genotype had no impact in the changes of the observed expression. 

### 2.2. Cosinor Analysis Revealed That Aging and NLRP3 Inflammasome Modified in Some Extend the Rhythm and Changes in the Acrophase of Clock Genes in Mice Heart—Melatonin Therapy Restored These Changes

The gene *Clock* presented an acrophase at 7.53 h in WT Y mice (Figure 3). This acrophase was significantly delayed in WT EA and OA mice. In both cases, melatonin counteracted this delay in the acrophase. A similar trend to that of WT mice was observed in NLRP3^−/−^ mice. The acrophase of *Clock* in Y mice averaged at 7.80 h. A phase delay was also observed with EA and OA mice. In both cases, melatonin corrected this phase change (Figure 3). Rhythm was appreciated in all experimental groups of WT and NLRP3^−/−^ mice (Appendix A). 

The acrophase of *Bmal1* showed some similarity to those of the *Clock* gene. *Bmal1* acrophase was slightly delayed in WT EA mice and it was corrected with melatonin (Figure 3). The acrophase of *Bmal1* also appeared at 8 h in NLRP3^−/−^ Y. Aging did not cause significant changes in the acrophase of *Bmal1* in mutant mice (Figure 3). Melatonin advanced the acrophase in both cases, placing it closer to that of the Y mice. The existence of rhythm was observed in all the experimental groups, both in WT and in NLRP3^−/−^ (Appendix A).

The *Per2* gene had an acrophase close to 20 h in WT Y mice, remaining constant with age. Interestingly, melatonin significantly advanced this acrophase at 12 and 24 months (Figure 3). *Per2* lost its rhythm in WT OA mice, which was restored with melatonin (Appendix A). No prominent changes were observed with age or with melatonin therapy in NLRP3^−/−^ mice (Figure 3). The acrophase was different with respect to the WT mice, occurring around 12 h. The rhythm was present in all experimental groups (Appendix A).

The acrophase of *Chrono* was around 20 h in the WT Y mice. No notable changes were observed with aging or with melatonin treatment (Figure 3). In mutant mice, the acrophase showed a trend similar to that observed in WT mice. In this case, the acrophase was found at around 18 h in Y mice, remaining constant throughout aging and with melatonin therapy (Figure 3). Mutant mice treated with melatonin showed a phase advanced in rhythm regarding WT. Rhythm was detected in all groups of WT and NLRP3^−/−^ (Appendix A).

The *Rev-erbα* acrophase remained constant during aging (Figure 3) in WT mice. Interestingly, melatonin produced a phase advance in EA and OA. The acrophase values for *Rev-erbα* in the NLRP3^−/−^ mice were similar to WT mice (Figure 3). With aging, there appeared to be a trend towards phase advancement. Melatonin had little effect on mutant mice. Again, there was rhythm in all experimental groups (Appendix A).

The *Rorα* acrophase occurred at nearly 12 h in WT Y mice (Figure 3). Rhythm was lost in WT EA and WT OA mice. Melatonin recovered the rhythm and advanced the acrophase by 5–6 h in both experimental groups (Appendix A). In NLRP3^−/−^ mice, the acrophase appeared at 18 h in Y mice (Figure 3). As in WT mice, the rhythm was lost at 12 and 24 months. Melatonin regained rhythm and produced a phase advance of an equivalent duration to that observed in WT mice (Appendix A).

### 2.3. Cosinor Analysis Revealed That Aging, the Absence of NLRP3 Inflammasome, and Melatonin Treatment Had Little Impact on the Amplitude of Clock Genes in Heart Muscle

The amplitude of the gene *Clock* remained constant with age and with melatonin treatment in WT mice (Figure 4). In mutant mice, the amplitude increased significantly in EA and in OA vs. Y mice. Melatonin decreased this amplitude to the values of Y group in NLRP3^−/−^ EA mice (Figure 4).

The amplitude of *Bmal1* was invariable with aging in WT mice (Figure 4). It decreased significantly in WT OA + aMT vs. WT OA mice. In NLRP3^−/−^ mice, the amplitude increased at 24 months and melatonin decreased the amplitude to the values of Y mice (Figure 4).

No changes were observed in the amplitude of the *Per2* and *Rev-erbα* gene in WT or NLRP3^−/−^ mice (Figure 4). 

The amplitude of *Chrono* increased in WT OA vs. WT Y. Melatonin counteracted this change (Figure 4). However, in the mutant mice, the amplitude decreased in NLRP3^−/−^ OA vs. NLRP3^−/−^ Y. Again, melatonin re-established the normal amplitude (Figure 4). Amplitude in mutant mice was higher than in WT mice.

The amplitude of the *Rorα* gene did not undergo significant changes with age or with melatonin treatment in WT mice (Figure 4). A decrease in the amplitude was observed in NLRP3^−/−^ OA vs. NLRP3^−/−^ Y mice, which was mended with melatonin (Figure 4).

### 2.4. Cosinor Analysis Showed That Aging Caused Significant Changes in Clock Genes’ Mesor, Generally Decreasing with Age—Treatment with Melatonin and the Absence of NLRP3 Had Little Effect on this Parameter

The mesor of *Clock* decreased with age in WT mice (Figure 5). Melatonin had no effect. In NLRP3^−/−^ mice, the mesor increased in the EA group and decreased in the OA (Figure 5). Melatonin had no effect.

In WT mice, mesor of *Bmal1* declined with age. Melatonin did not counteract this outcome (Figure 5). The mesor remained constant with aging in mutant mice. Melatonin decreased mesor in NLRP3-deficient mice (Figure 5).

The mesor of *Per2* decreased in WT OA vs. WT Y mice. No effect of melatonin was observed (Figure 5). In NLRP3^−/−^ mice, a decrease in the mesor was again seen in the 24-month-old mice, although in this case melatonin restored the mesor to similar values to those of Y mice (Figure 5).

The mesor of *Chrono* remained constant during aging in WT mice (Figure 5). Interestingly, melatonin significantly decreased this mesor. The mesor diminished with age in NLRP3^−/−^ (Figure 5). Melatonin had no effect in mutant mice. In general, the mesor had higher values in NLRP3^−/−^ vs. WT mice.

No changes were observed in the mesor of the *Rev-erbα* gene derived from aging, treatment with melatonin or absence of the NLRP3 inflammasome (Figure 5).

The mesor of *Rorα* increased in WT OA vs. WT Y mice (Figure 5). Melatonin decreased the mesor significantly in the OA group. In contrast, the mesor was decreased in NLRP3^−/−^ OA vs. NLRP3^−/−^ Y mice (Figure 5). Melatonin had no effect in mutant mice. Again, higher mesor values were seen in NLRP3-deficient mice compared to WT mice.

### 2.5. Aging-Induced Morphological Alterations of Cardiomyocytes: Melatonin Supplementation and Lack of NLRP3 Inflammasome Conserved Cardiac Muscles Structure with Aging

Using light microscopy, left ventricles of young WT animals revealed a normal cardiac structure. Branched cardiomyocytes with centrally positioned nuclei attached at the intercalated disc were observed. The myocytes were divided by narrow interstitial spaces containing blood capillaries and fibroblasts. Furthermore, cardiac fibers illustrated longitudinal and transverse striations (Figure 6a). Cardiomyocyte striations of EA mice were conserved and demonstrated widening of interstitial spaces (Figure 6b). The cardiomyocytes of OA animals depicted progressive alterations, where the myocardium showed disorganization of cardiac fibers, loss of striation, and splitting of the nucleus (Figure 6d). Melatonin administration, however, elucidated a protective effect on the cardiac muscle fibers of both EA (Figure 6c) and OA (Figure 6e) WT mice. The fibers conserved their normal architecture with narrow interstitial spaces and an absence of necrotic damage.

Histological examination of heart tissue from Y NLRP3^−/−^ mice (Figure 6f) showed a normal organization of cardiomyocytes, which also demonstrated no changes in the EA animals (Figure 6g). Increased interstitial space and blood capillaries were observed in OA mutants (Figure 6i) compared with those of WT mice. Interestingly, melatonin therapy induced a more preservative effect on the cardiomyocytes of EA (Figure 6h) and OA (Figure 6j) NLRP3^−/−^ mice than in WT mice, keeping normal cardiomyocyte architecture with proper orientation and striation, and narrow interstitium containing blood capillaries.

## 3. Discussion

Our study provided worthwhile and novel findings that outline simultaneously, for the first time to the best of our knowledge, the effect of aging, NLRP3 inflammasome, and melatonin therapy on primary and secondary clock-loop components in cardiac tissue (Figure 7). The results of the present analysis yielded three main new discoveries: (1) aging, melatonin, and NLRP3 inflammasome had significant effects on clock gene expression in heart tissue, except for *Rev-erbα*, which was not affected by mice genotype; (2) aging and NLRP3 inflammasome affected the clock gene circadian rhythm in the heart. Aging caused small phase changes in *Clock*, loss of rhythmicity in *Per2* and *Rorα*, and a tendency towards dampening the mesor. NLRP3 inflammasome influenced the acrophase of *Clock*, *Per2*, and *Rorα*; (3) melatonin restored the acrophase and rhythm of these three genes in cardiac tissue, having clinical interest in managing heart chronodisruption. 

Aging and melatonin treatment influenced circadian clock gene expression in cardiac tissue. We found that ablation of NLRP3 inflammasome altered daily expression patterns of all clock genes investigated, excepting *Rev-erbα*. *Rev-erbα* exerts proinflammatory actions by binding competitively with RORα to the same RORE in the promoter of *Bmal1* [10], reducing its expression. Among other processes, BMAL1 regulates *Nampt* expression, whose protein synthesizes NAD^+^, the cofactor of SIRT1. This deacetylase inactivates NF-κB, therefore controlling the inflammatory response [34]. Additionally, the anti-inflammatory effect of REV-ERBα also downregulates the expression of Nlrp3 [34,39,40,41]. On the other hand, REV-ERBα seems to have anti-inflammatory actions which may be tissue-specific [39,42]. Most studies reveal a protective role of REV-ERBα in cardiac tissue by inhibiting atherosclerosis risk factors [42,43,44,45,46] and downregulation of NLRP3 inflammasome [47,48,49]. The molecular mechanism of this anti-inflammatory effect remains poorly understood in hearts. Using heart tissue ChIP-Seq, a recent study proposed that *Rev-erbα* can colocalize with other transcription factors and coordinate the repression at thousands of loci mediated by multiple transcription factors, preventing pathogenic genetic development [49]. Recent studies showed increased *Rev-erbα* in murine cardiac tissue with age [35]. Our results, however, illustrated that the absence of *Nlrp3* did not alter the mRNA expression of *Rev-erbα*, and the presence of *Nlrp3* did not appear to influence *Rev-erbα* in murine cardiac tissue.

The core components of the circadian clock, *Clock* and *Bmal1*, are vital for cardiac physiology. Global and cardiomyocyte-specific mutant mice for *Clock* and *Bmal1* genes leads to premature aging and the development of age-associated cardiomyopathy [50,51,52,53,54]. Cosinor analysis reflected that the rhythms of *Clock* and *Bmal1* were not affected by aging. Some authors did not observe the rhythm of *Clock* in the hearts of young Balb/c mice [55]. However, our findings are in line with those who showed *Clock* rhythmicity and similar acrophase to *Bmal1* in young age rodents [56,57]. Although the rhythm of gene *Clock* in the heart was unaffected, the phase was delayed over 4 h in WT EA and OA vs. WT Y mice, which agrees with that observed in other tissues [58,59]. Rhythms and acrophases of *Bmal1* were preserved at all ages, as found in human skin fibroblasts, cortical area, and mouse brain and liver [60,61,62]. This fact is important for the maintenance of the circadian rhythm in the heart since *Bmal1* is the only obligate mammalian clock gene for rhythmicity [63]. Absence of *Nlrp3* preserved the acrophase of *Clock* and *Bmal1* in almost all ages, indicating the influence of this inflammasome on age-related acrophase shifts. Melatonin corrected the alterations in *Clock* acrophases, probably by counteracting the disruption of CLOCK/BMAL1/NF-κB/SIRT1 in aged mice and reducing NLRP3 inflammasome activation [24,35]. 

Our results showed that *Bmal1* is expressed in antiphase with the *Per2* gene in WT Y and EA, coinciding with prior studies in heart and other peripheral tissues [55,64]. The acrophase is maintained at the beginning of the dark phase (20–21 h), but rhythm is lost with aging in WT OA animals. Although some studies suggested the absence of changes in *Per2* resulting from the course of aging [61,65], our data agreed with other reports that revealed consistent effects in the *Per2* circadian pattern of aged animals [66,67,68,69]. Studies examining the role of *Per2* in cardiac function were conflicting. Some authors found *Per2*-mutant mice had less severe injury in ischemia/reperfusion and nonreperfused myocardial infarction than control mice [70,71]. Conversely, the cardioprotective effect of *Per2* as a mediator of endothelial function, vascular senescence, and angiogenesis was established [72,73,74]. Melatonin restored the rhythm in WT OA mice and significantly advanced the acrophase in both WT EA and WT OA mice. The phase advance of *Per2* in control and hypertensive rat hearts was previously described after melatonin administration in drinking water during the dark phase for 6 weeks [75]. Melatonin phase advance may be cardioprotective by extending the period, whose length decreases as a result of aging [76,77,78,79], by inhibiting PER2 proteasomal degradation [80]. IL-1β, which is triggered as a result of NLRP3 inflammasome activation, disrupts the circadian rhythm of Per2 in peripheral tissues [81]. The absence of NLRP3 maintained the rhythm in OA mice, possibly because IL-1β is less active without the action of the inflammasome. The aged-associated increase of NLRP3 may cause the loss of rhythm in WT OA mice. The acrophase of mutant mice remained constant with age and melatonin treatment. Interestingly, this acrophase was notably different from WT. This fact suggests that inflammaging caused by NLRP3 activity may induce *Per2* acrophase impairment.

Similarly to the *Per2* gene, the acrophase of *Chrono* occurred at 20 h in WT Y mice. Both genes act as repressors of the *Clock*/*Bmal1* positive loop and are in antiphase with *Bmal1* [82,83]. However, unlike *Per2*, the acrophase and rhythmicity of *Chrono* remained constant with aging and with melatonin therapy in WT mice. These data suggest that *Chrono* is an evolutionarily highly preserved gene. As a matter of fact, *Chrono* is known to be the gene that is rhythmically expressed in the largest number of tissues in diurnal primates [84]. Lack of NLRP3 had no impact on the rhythm and acrophase of *Chrono*, which were similar to WT observations. This data implies that, contrary to *Per2*, *Nlrp3* expression does not influence *Chrono* circadian activity. Overall, mechanisms of action and regulation of *Per2* and *Chrono* seem to be different. This result is in line with recent discoveries that found that *Per2* and *Chrono* bind *Bmal1* at N- and C-terminus, respectively, and act distinctly as repressors in the mammalian circadian clock [85,86].

The rhythm and acrophase of *Rev-erbα* persisted in the aged heart of WT mice [55], and melatonin induced phase advance of *Rev-erbα* in WT EA and OA mice, suggesting that the chronobiotic effect of melatonin may rely on *Rev-erbα* as an initial molecular target [44]. Aging caused phase advance in mutant mice as it was observed in WT melatonin-treated mice. It appears that the therapeutic effect of melatonin on *Rev-erbα* acrophase is mimicked in NLRP3^−/−^ mice, corroborating that some protective effects of melatonin in WT cardiac tissue is dependent on the suppression of NLRP3 inflammasome [24,35].

The *Rorα* rhythm disappeared in the hearts of WT EA/OA mice. Conversely, neither rhythm nor systematic changes were found in *Rorα* expression of gastrointestinal tissues with aging [87]. Studies related to the age-associated changes in *Rorα* are very limited, suggesting a tissue-specific function. This gene is considered a molecular link between circadian rhythm and cardiac homeostasis [88]. Mice with a loss-of-function mutation in RORα (*Rora*^sg/sg^) have impairments in the circadian oscillator and develop severe cardiomyopathies. Pharmacological activation of RORα ameliorated the deleterious cardiac changes and strengthened circadian oscillations [89,90]. In this sense, melatonin recovered the rhythm of this gene in WT EA and OA animals and, interestingly, advanced the phase of *Rorα* before that of *Bmal1*; this response may enhance its anti-inflammatory action during aging, also observed in the cardiac sepsis mice model [35]. *RORα* is a known activator of the BMAL1/NAD+/SIRT1 anti-inflammatory pathway. Melatonin seems to modulate age-related inflammatory response through *RORα* [11,34,91]. Lack of NLRP3 shifted the phase at 18 h in Y mice with less rhythmicity compared to WT Y animals, indicating that NLRP3 may have a role in maintaining the circadian rhythm in the heart [11]. Melatonin re-established *Rorα* rhythm and advanced the acrophase to a similar degree in both strains of mice, probably because there are no changes in *Nampt* and *Sirt1* expression in the mutant mice [92], limiting the activation of *Rorα* in Y mice. Thus, *Rorα* rhythm and acrophase are influenced by melatonin and NLRP3 inflammasome. 

Lower amplitude and diminished mesor have been linked with an augmented risk of CVDs [93]. Aging reduced the amplitude in both central and peripheral tissues of mammals and *Drosophila melanogaster* [66,94,95,96]. However, a significant number of studies showed no apparent effect of aging regarding these circadian parameters [61,62,97,98]. Our results suggest that aging, lack of NLRP3 inflammasome, and melatonin treatment had low influence in the amplitude of clock genes in the heart. Instead, NLRP3 absence and melatonin did not restore the mesor, which tended to decline with aging. Contrary to our findings, Bonaconsa et al. found a tendency towards amplitude decrease and preservation of the mesor in aged hearts [55]. Controversial results regarding amplitude have been observed in rodent SCN, human leukocytes, mucosa and heart, cardiac tissue being the one with the widest range of amplitudes [99,100]. The intense metabolism and low rate of differentiation of myocardial cells were proposed by authors as a possible explanation of these results [101,102].

Light microscopy analysis of cardiomyocytes revealed age-associated morphological changes, where fibers lost their organization with age, accompanied with a widening of interstitial spaces. Moreover, age induced a lack of muscle fiber striations and splitting of nuclei. These alterations with age were less prominent in NLRP3^−/−^ mice than WT mice. Melatonin supplementation preserved cardiomyocyte structure in all experimental groups. Our previous studies confirmed that cardiac sarcopenia begins at the age of 12 months with a wide variety of morphological impairments, including cardiac hypertrophy, reduced cardiomyocyte number, destroyed mitochondrial cristae, and increased apoptotic nuclei ratio, with the appearance of small necrotic fibers and small-sized residual bodies. These structural changes were exacerbated in the old-aged animals, accompanied by mitochondrial vacuolization and damage, excessive collagen deposition, myofibril disorganization, and formation of multivesicular bodies. All morphological alterations were restored with melatonin supplementation, preserving normal cardiac architecture [23,24]. Even though it is widely accepted that circadian disruption is implicated in heart diseases, it is unclear whether cardiac molecular clock dysfunction is associated with the observed morphological changes as well of the occurrence of human heart diseases [103]. Recent investigations showed the relevance of cardiac clock genes in the cardiovascular physiology. Cardiomyocyte-specific Bmal1 deletion in mice resulted in diastolic dysfunction and fibrosis [52]. Loss of *Clock* gene activity affected mitochondria dynamics and mitophagy in cardiomyocytes during ischemic stress [104], resembling our results obtained in OA mice in previous studies [23,24]. Similarly, melatonin had a protective effect in the structure and function of the heart and re-established the rhythm of cardiac clock genes. Taking all the data together, it suggests that there is a connection between the expression pattern changes in cardiac clock genes and the impairments in the architecture and function of cardiomyocytes. However, further studies, and particularly those focused on specific clock gene-knockout in hearts, are needed to elucidate this incognita. 

In summary, although we observed changes in the parameters of cardiac circadian rhythm, the described alterations were, in general, lower in magnitude relative to other organs, including SCN or liver [55,105,106]. This suggests a tissue-specific function in circadian rhythms and a propensity for preservation of heart chronobiology. In concordance with our previous findings, sarcopenia was more severe in the skeletal muscle than the cardiac tissue [23,24]. Melatonin also improved the rhythmicity in cardiac muscle, having a clinical interest in preventing and treating chronodisruption and sarcopenia induced by inflammaging.

## 4. Materials and Methods

### 4.1. Animals

Wild-type C57BL/6J (RRID:IMSR JAX:000664) and NLRP3-knockout mice NLRP3^−/−^ (B6.129S6-NLRP3*^tm1Bhk^*/J) (RRID:IMSR_JAX:021302) on the wild-type C57BL/6J background (>10 backcrosses) aged 3 weeks, were purchased from The Jackson Laboratory (Bar Harbor, ME, USA). Animals were housed in the animal facility of the University of Granada under a specific pathogen-free barrier facility and conditions of constant temperature (22 °C ± 1 °C), controlled 12 h light/dark cycle (lights on at 08:00 h), and free access to tap water and rodent chow. Mice were distributed into cages with a maximum of 3 mice in each. 

Mice were treated in accordance with the National Institutes of Health Guide for the Care and Use of Laboratory Animals, the European Convention for the Protection of Vertebrate Animals used for Experimental and Other Scientific Purposes (CETS # 123), and the Spanish law for animal experimentation (R.D. 53/2013). The protocol was authorized by the Andalusian’s Ethical Committee (#05/07/2016/130).

Wild-type (WT) and NLRP3^−/−^ male and female mice were classified into five experimental groups regardless of gender (n = 6 animals per group): (I) young (Y, 3-months old), (II) early-aged (EA, 12-months old), (III) early-aged plus melatonin (EA + aMT), (IV) old-aged (OA, 24-months old), and (V) old-aged plus melatonin (OA + aMT) mice. A synthetically sourced melatonin (Cat# 33457-24, Fagron, Barcelona, Spain) was orally administered at 10 mg/kg/day in the food during the last two months before early and old-aged treated mice were sacrificed (EA + aMT at the age of 10 months and OA + aMT at the age of 22 months). The integration of melatonin to the chow’s pellet was performed in the Diet Production Unit facility of the University of Granada. With the purpose of ensuring that each mouse received 10 mg/kg body weight daily, the amount of melatonin in the pellets was prepared according to average daily food intake, number, weight, and age of mice. A sample of melatonin chow was analyzed by HPLC as a quality control (data not shown). The other experimental groups (Y, EA, and OA) were fed with the same chow without melatonin. This dosing regimen of melatonin, given during day and night, was selected on the basis of its effectiveness on previous experimental models, among them prevention of aging in SAM mice [107,108], as well as sarcopenia and inflammaging in murine skeletal and cardiac muscle of C57/BL6J mice [23,24,109,110]. Some studies have reported that C57/BL6 mice are melatonin-deficient mice [111,112]. Nevertheless, we and other researchers have demonstrated that they produce pineal and extrapineal melatonin. Moreover, C57/BL6 mice respond well to melatonin treatment [113,114]. Therefore, this mice strain is appropriate for the purpose of this investigation.

To study the circadian rhythm, mice were sacrificed by cervical dislocation after equithensin anesthesia via intraperitoneal injection (1 mL/kg), at 24:00, 06:00, 12:00, and 18:00 h under a light/dark cycle. Hearts were collected, washed in cold saline, and freshly stored at −80 °C for PCR analysis. During the night, animals were sacrificed under a dim red light to avoid influence on endogenous melatonin production [115].

### 4.2. Real-Time Reverse Transcription Polymerase Chain Reaction (RT-PCR)

RNA was isolated from frozen mouse hearts using the NZY Total RNA Isolation kit (Cat# MB13402, Nzytech Genes and Enzymes, Lisbon, Portugal). An initial proteinase K digestion step was performed to improve the yield of RNA (20 mg/mL proteinase K, 600 mAU/mL) (Cat# 191133, Qiagen, Hilden, Germany). RNA was quantified in a Nano Drop ND-1000 spectrophotometer (Thermo Scientific, Wilmington, DE, USA), and its integrity was confirmed by 2% agarose gel electrophoresis. RNA was reverse transcribed to cDNA with a qScript^TM^ cDNA synthesis kit (Cat# 95047, Quanta Biosciences, Gaithersburg, MD, USA). Amplification was performed by quantitative real-time polymerase chain reaction (RT-PCR) in a Stratagene Mx3005P QPCR System (Agilent Technologies, Madrid, Spain) with SYBR^®^ Premix Ex Taq^TM^ (Cat# RR420, Takara Bio Europe, Saint-Germain-en-Laye, France). Primer sequences (Appendix A) were designed using the Beacon Designer^TM^ Software (Premier Biosoft, Palo Alto, CA, USA). The thermal profile of RT-PCR was as follows: 10 min at 96 °C before 40 thermal cycles, each consisting of 15 s at 95 °C and 1 min at 55 °C. Output data were analyzed with the MxPro QPCR Software v 4.0 (Agilent Technologies) according to the standard curves generated from increasing amounts of cDNA (0.05, 0.5, 5, 50, and 500 ng). The beta-actin housekeeping gene was used as an endogenous reference gene. Template-free (water) sample was used as a negative control and 3 months-old wild type mice were used as a calibrator sample.

### 4.3. Quantification and Statistical Analysis

The quantity of mRNA was expressed as mean ± standard error of the mean (SEM) and was generated in triplicate RT-PCR amplifications. Data were normalized as the percentage of the highest recorded value. Cosinor analysis [116] was performed with the Time Series Analysis-Seriel Cosinor 6.3 Lab View software (TSASC 6.3; Expert Soft Technologies Inc, BioMedical Computing and Applied Statistics Laboratory, Esvres, France). Rhythm characterization included the average level of the acrophase, amplitude, and mesor, calculated with 95% confidence limits. Rhythm detection was considered statistically significant at *p* < 0.05. To evaluate the effect of the time-point of mice sacrifice, age, melatonin treatment, and genotype in the expression of clock gene transcripts, multifactorial analysis of variance (ANOVA) analysis was performed with R software v 4.0.2. Significant differences were evaluated by Tukey post hoc test.

ANOVA tests the hypothesis that the means of two or more populations are equal. ANOVAs assess the importance of one or more factors by comparing the means of the response variable at different levels of the factors. The null hypothesis establishes that all the population means (means of the factor levels) are equal while the alternative hypothesis establishes that at least one is different. The ANOVA requires a continuous response variable and at least one categorical factor with two or more levels, as in our case. ANOVA analyses require population data that follow an approximately normal distribution with equal variances between factor levels. For the adjustment of the multifactorial ANOVA model, the characteristics of the data were analyzed. In each case, the required hypotheses were verified: each unit was independent of the other; individual measurements were obtained from each individual; the effect of the predictor variables (factors, nominal variables) was analyzed, and data were available for any combination of factors.

After estimating the ANOVA models, the residuals were analyzed. It was observed that the hypotheses of normality (considering original data or Box-Cox transformations), homoscedasticity, and no correlation were verified. With a significance level of 5%, the hypotheses of normality were verified with the Shapiro–Wilks test. The lack of correlation in the residuals was verified with the Durbin–Watson test. Graphical analysis was used to analyze homoscedasticity. 

### 4.4. Light Microscopy Analysis of Cardiac Muscle Fibers

To assess the effect of aging, NLRP3 inflammasome deletion, and melatonin supplementation on the architecture of cardiac muscle fibers, five mice from each experimental group were anesthetized using ketamine and xylazine, and were transcardially perfused with warm saline, followed by a freshly prepared trump’s fixative [117]. Fresh pieces of the left ventricle were fixed in 2.5% glutaraldehyde in 0.1 M cacodylate buffer, and post fixed in the former buffer with 1% osmium tetraoxide plus 1% potassium ferrocyanide. The samples were then immersed in 0.15% tannic acid, incubated in 1% uranyl acetate, dehydrated in ethanol, and embedded in resin. Semithin sections of 0.65 μm thickness were cut by a Reichert-Jung Ultracut E ultramicrotome and stained with 1% toluidine blue for light microscope analysis [118]. The sections were examined using a Carl Zeiss Primo Star Optic microscope, and digital images were acquired using a Magnifier AxioCam ICc3 digital camera (BioSciences, Jena, Germany). 

## Figures and Tables

**Figure 1 ijms-23-06846-f001:**
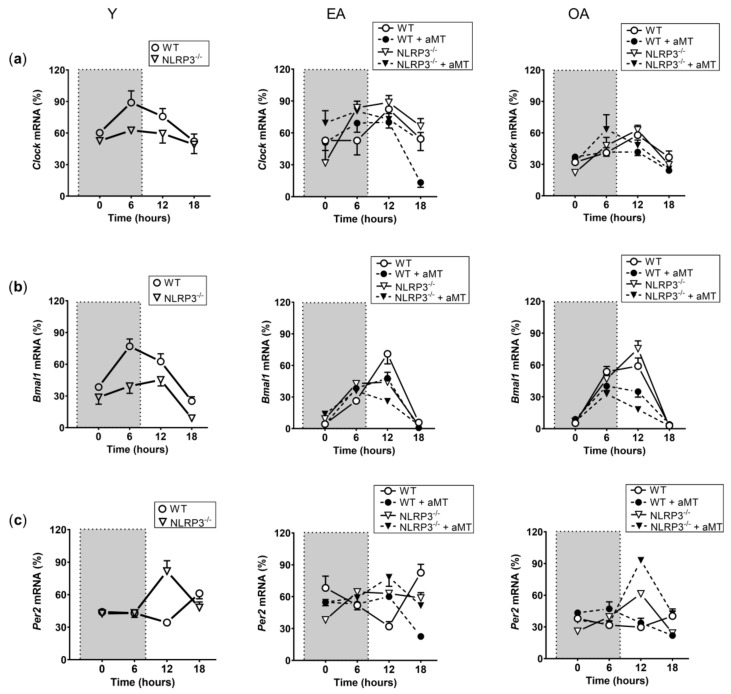
Changes in the relative expression of the *Clock*, *Bmal1*, and *Per2* transcripts in WT and NLRP3^−/−^ mice during aging and melatonin treatment. Relative expression of the *Clock* (**a**), *Bmal1* (**b**), and *Per2* (**c**) transcripts in hearts of young (Y), early-aged (EA), early-aged with melatonin (EA + aMT), old-aged (OA), and old-aged with melatonin (OA + aMT) wild type and NLRP3^−/−^ mice and their daily patterns at 4 time points (00:00, 06:00, 12:00, and 18:00 h) under a 12 h/12 h light/dark cycle. The shaded region on each graph represents constant darkness. Data are expressed as means ± SEM (n = 6 animals/group). Data of significant differences among the experimental groups were evaluated by *Tukey multiple comparison* of mean post hoc test following ANOVA, and they are detailed in Appendix A: Appendix A (*Clock*), Appendix A (*Bmal1*), Appendix A (*Per2*).

**Figure 2 ijms-23-06846-f002:**
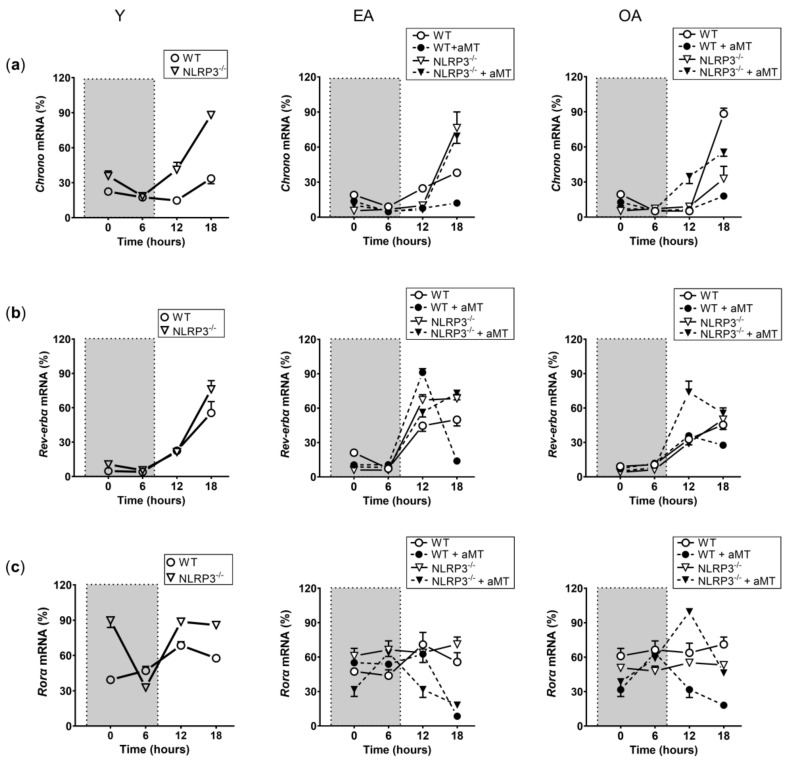
Changes in the relative expression of the *Chrono*, *Rev-erbα*, and *Rorα* transcripts in WT and NLRP3^−/−^ mice during aging and melatonin treatment. Relative expression of the *Chrono* (**a**), *Rev-erbα* (**b**), and *Rorα* (**c**) transcripts in hearts of young (Y), early-aged (EA), early-aged with melatonin (EA + aMT), old-aged (OA), and old-aged with melatonin (OA + aMT) wild type and NLRP3^−/−^ mice and their daily patterns at 4 time points (00:00, 06:00, 12:00, and 18:00 h) under a 12 h/12 h light/dark cycle. The shaded region on each graph represents constant darkness. Data are expressed as means ± SEM (n = 6 animals/group). Data of significant differences among the experimental groups were evaluated by *Tukey multiple comparison* of mean post hoc test following ANOVA and are detailed in Appendix A: Appendix A.4 (*Chrono*), S1.5 (*Rev-erbα*), S1.6 (*Rorα*).

**Figure 3 ijms-23-06846-f003:**
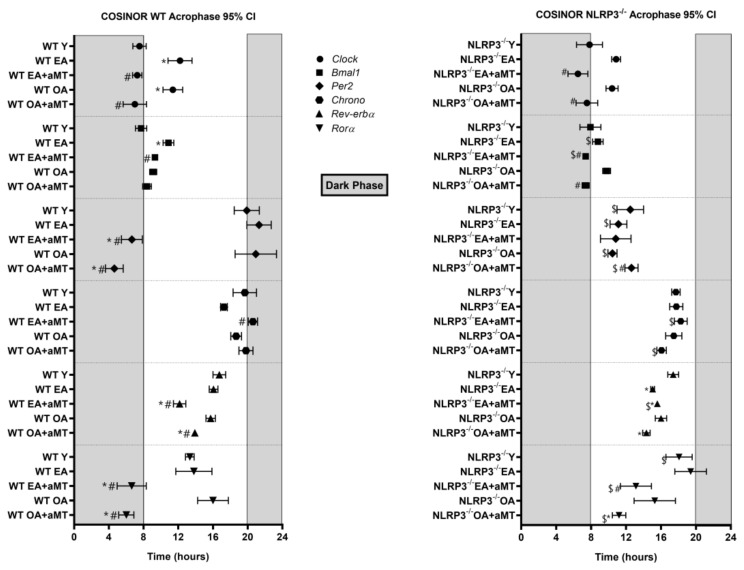
Acrophase charts showing peaks of fitted 24 h cosine for clock genes analyzed in the heart of WT and NLRP3^−/−^ mice during aging and melatonin treatment. The acrophase of clock genes *Clock*, *Bmal1*, *Per2*, *Chrono*, *Rev-erbα*, and *Rorα* in hearts of young (Y), early-aged (EA), early-aged with melatonin (EA + aMT), old-aged (OA), and old-aged with melatonin (OA + aMT) wild type (left chart) and NLRP3^−/−^ mice (right chart). The shaded region on each graph represents constant darkness. Data are expressed as means ± confidence interval (CI) of the acrophase when *p* was ≤0.05 from the nonzero amplitude. Data of significant differences among the experimental groups were evaluated by overlapping of confidence interval and are detailed in Appendix A. * *p* < 0.05 vs. Y; # *p* < 0.05 vs. group without melatonin treatment; $ *p* < 0.05 vs. WT mice.

**Figure 4 ijms-23-06846-f004:**
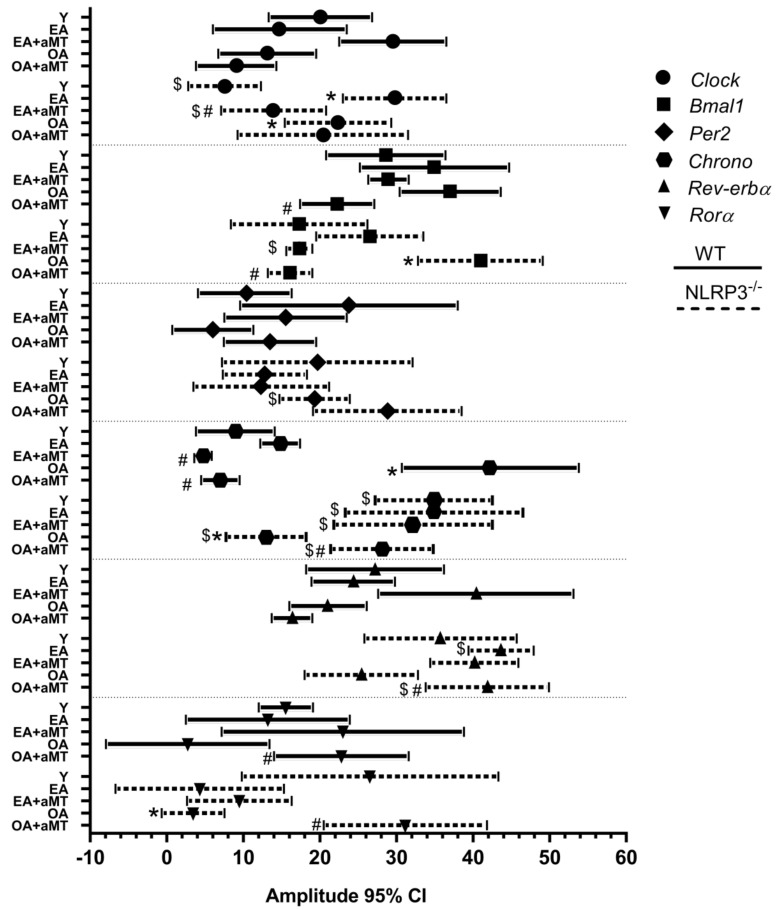
Amplitude chart showing peaks of fitted 24 h cosine for clock genes analyzed in the heart of WT and NLRP3^−/−^ mice during aging and melatonin treatment. Amplitude of clock genes *Clock*, *Bmal1*, *Per2*, *Chrono*, *Rev-erbα*, and *Rorα* in hearts of young (Y), early-aged (EA), early-aged with melatonin (EA + aMT), old-aged (OA), and old-aged with melatonin (OA + aMT) wild type (solid line) and NLRP3^−/−^ mice (dashed line). Data are expressed as means ± confidence interval (CI) of the amplitude when *p* was ≤ 0.05 from the nonzero amplitude. Data of significant differences among the experimental groups were evaluated by overlapping of confidence interval and are detailed in Appendix A. * *p* < 0.05 vs. Y; # *p* < 0.05 vs. group without melatonin treatment; $ *p* < 0.05 vs. WT mice.

**Figure 5 ijms-23-06846-f005:**
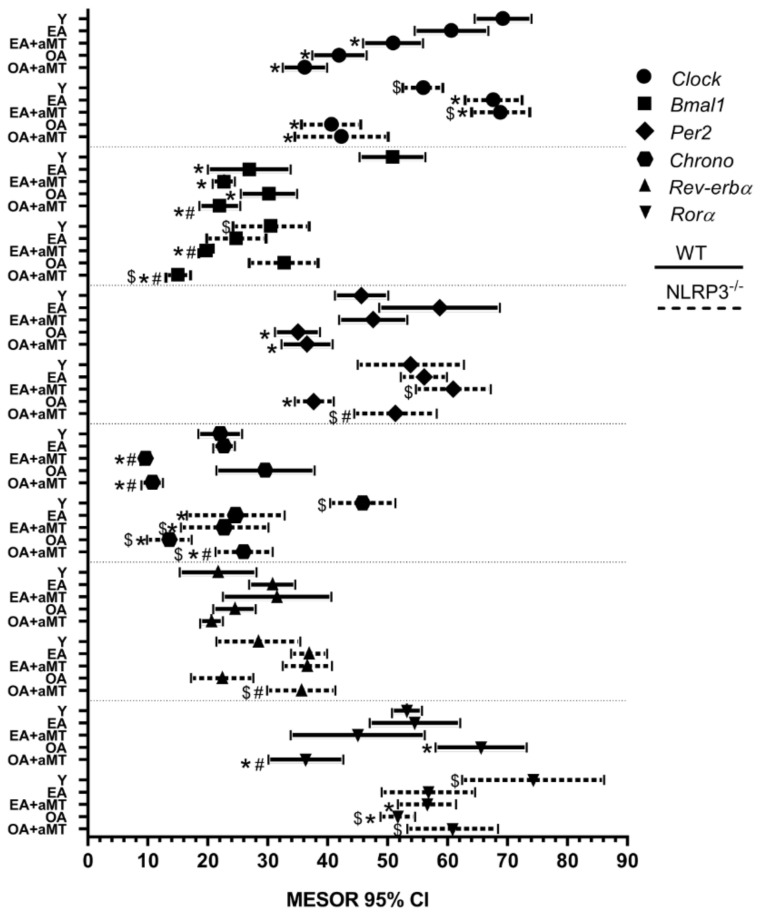
Mesor chart showing peaks of fitted 24 h cosine for clock genes analyzed in the heart of WT and NLRP3^−/−^ mice during aging and melatonin treatment. The mesor of clock genes *Clock*, *Bmal1*, *Per2*, *Chrono*, *Rev-erbα*, and *Rorα* in hearts of young (Y), early-aged (EA), early-aged with melatonin (EA + aMT), old-aged (OA), and old-aged with melatonin (OA + aMT) wild type (solid line) and NLRP3^−/−^ mice (dashed line). Data are expressed as means ± confidence interval (CI) of the mesor when *p* was ≤ 0.05 from the nonzero amplitude. Data of significant differences among the experimental groups were evaluated by overlapping of confidence interval and are detailed in Appendix A. * *p* < 0.05 vs. Y; # *p* < 0.05 vs. group without melatonin treatment; $ *p* < 0.05 vs. WT mice.

**Figure 6 ijms-23-06846-f006:**
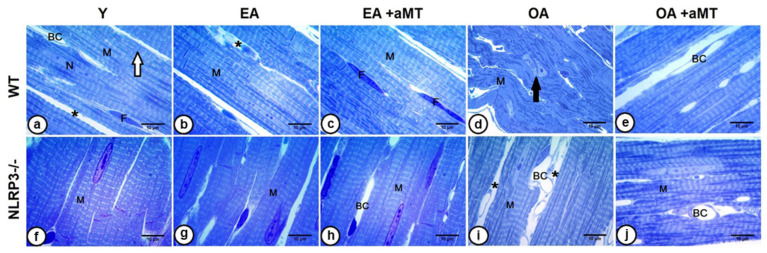
Effect of aging, NLRP3 deletion, and melatonin supplementation on morphological architecture of cardiac muscle fibers. (**a**) Semithin left ventricle sections of young (Y) WT mice stained by toluidine blue stain showing normal cardiac muscle fiber (M) architecture, with centrally positioned nucleus (N), and narrow interstitial spaces (asterisk) containing blood capillaries (BC) and fibroblasts (F). Note the intercalated disc (white arrow). (**b**) Cardiomyocytes (M) of early-aged (EA) WT mice revealing conservation of their striations with widening of interstitial spaces (asterisk). (**c**) The preservative effect of melatonin on maintaining normal cardiac fiber (M) architecture in EA animals. Numerous fibroblasts (F) were shown. (**d**) Cardiac muscle fibers of old-aged (OA) WT mice demonstrating disorganization of cardiomyocytes (M), loss of striation, and splitting of the nucleus (black arrow). (**e**) The protective effect of melatonin on conserving cardiac muscle fiber (M) orientation and striation in OA animals. (**f**) Left ventricle of Y NLRP3^−/−^ mice showing normal organization of the cardiomyocytes (M). (**g**,**i**) Cardiac muscle fibers (M) of the EA and OA NLRP3^−/−^ mice, respectively, revealing no structural changes, except increased interstitial tissues (asterisks) and blood capillaries (BC) in OA muscles. (**h**,**j**) Depicts the preservative effect of melatonin on cardiomyocyte (M) architecture in EA and OA NLRP3^−/−^ mice, respectively, revealing normal orientation and striation with narrow interstitial space containing blood capillaries (BC). Bar = 10 µm.

**Figure 7 ijms-23-06846-f007:**
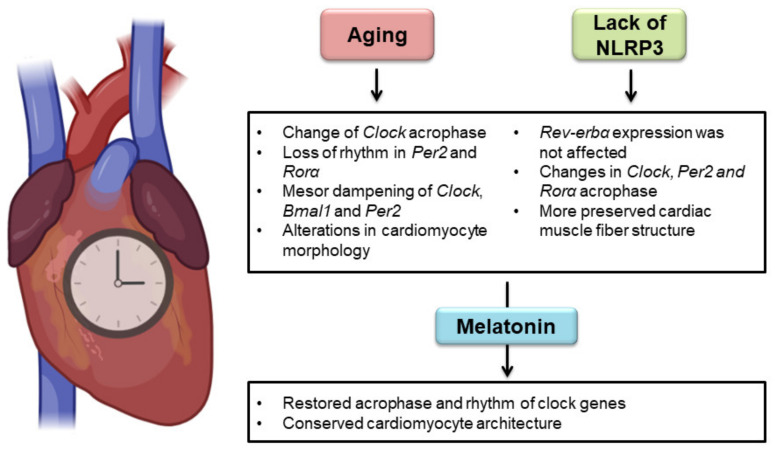
Effect of aging, NLPR3 inflammasome, and melatonin therapy on the cardiac circadian system. Aging produced minor changes in the *Clock* acrophase, loss of rhythmicity in *Per2* and *Rorα*, and a tendency towards dampening the mesor. NLRP3 inflammasome modified the acrophase of *Clock*, *Per2*, and *Rorα*. Expression of *Rev-erbα* was not influenced by mouse genotype. Melatonin re-established the acrophase and rhythm of these three genes. Morphological analysis revealed age-associated impairments in cardiac muscle fibers, which were less harmful in NLRP3-knockout mice. Melatonin treatment preserved the structure of cardiomyocytes in all experimental groups.

**Table 1 ijms-23-06846-t001:** Main effect of time point, age, and genotype, and their interaction in *Clock*, *Bmal1*, *Per2*, *Chrono*, *Rev-erbα*, and *Rorα* transcripts. The F and *p* values for the main effect of time point, age, and genotype, and their interaction for each gene were analyzed by Multifactorial-ANOVA. A significant effect was considered when *p* value < 0.05.

Multifactorial Anova Analysis								
Transcript	Main Effect of Time Point		Main Effect of Age		Main Effect of Genotype		Time Point × Age × Genotype Interaction	
	F _(3,59)_	*p* Value	F _(4,59)_	*p* Value	F _(1,59)_	*p* Value	F _(12,59)_	*p* Value
** *Clock* **	81.80	<0.001	47.60	<0.001	9.87	<0.01	6.33	<0.001
** *Bmal1* **	781.11	<0.001	83.88	<0.001	9.14	<0.05	8.37	<0.001
** *Per* **	23.31	<0.001	38.11	<0.001	42.17	<0.001	10.34	<0.001
** *Chrono* **	744.15	<0.001	179.12	<0.001	33.91	<0.001	34.34	<0.001
** *Rev-erbα* **	342.37	<0.001	8.38	<0.001	No effect		8.02	<0.001
** *Rorα* **	9.51	<0.001	16.34	<0.001	19.92	<0.001	14.02	<0.001

## Data Availability

The datasets generated during and/or analyzed during the current study are available from the corresponding author (dacuna@ugr.es) on reasonable request. Materials described in the manuscript will be freely available to any research to use them for noncommercial purposes.

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
