# Peer review of "Age and Chronodisruption in Mouse Heart: Effect of the NLRP3 Inflammasome and Melatonin Therapy"

_ijms, 2022, doi:10.3390/ijms23126846_

Round 1

Reviewer 1 Report

While there is value in the study, there are major issues relating to the design and interpretation of the study, as well as the significance of the altered expression patterns of several clock genes under different conditions (aging, melatonin, NLRP3-/-). The described effects are inconsistent with the actual data, confusing and not very informative. Below are some but not an exhaustive list of concerns.

1.     The time course was collected during the light/dark cycle and therefore does not inform circadian gene expression. The time course was at a 6 h interval, which does not allow for the accurate discerning of acrophases. For example, Chrono and Per2 are not known to peak at the same time. The acrophase of Rora is doubtful.  

2.     The conclusions and statements are often inconsistent, at times contradictory, about the effects of aging, melatonin, and NLRP3 deletion on several of the core clock genes. This is related to the poor design of the time course and poor data quality.

3.     It is not clear how the expression pattern changes of these genes impact cardiovascular physiology.

4.     There is no data presented that validates the NLRP3-/- model.

5.     An old reference #26 (Belle et al, Cell Cycle, 2012) was used to support the statement that “The influence of NF-kB on the disruption of circadian rhythms during aging is well-established”. However, there have been several more recent papers, for example, Haspel et al, Nat Commun, 2014; Hong et al, G&D, 2018; and Shen et al, PLoS Genet, 2021.

Author Response

While there is value in the study, there are major issues relating to the design and interpretation of the study, as well as the significance of the altered expression patterns of several clock genes under different conditions (aging, melatonin, NLRP3-/-). The described effects are inconsistent with the actual data, confusing and not very informative. Below are some but not an exhaustive list of concerns.

 An extensive list of articles was read to design our study and interpret our results carefully. As pointed out by this reviewer, one of the main challenges regarding clock genes are its complexity, the lack of information and the contradictory nature of published data, as expressed in our Discussion section and also reported by Banks et al, 2016. We hope our manuscript can shed more light on this topic and contribute to a better understanding of the effects that aging, NLRP3 inflammasome and melatonin therapy have on clock genes in mouse heart. To our knowledge, this is the first time that all of these conditions are addressed in one single study.

 Reference:

  • Banks, G.; Nolan, P.M.; Peirson, S.N. Reciprocal Interactions between Circadian Clocks and Aging. Mamm Genome 2016, 27, 332–340, doi:10.1007/s00335-016-9639-6.

  1. The time course was collected during the light/dark cycle and therefore does not inform circadian gene expression. The time course was at a 6 h interval, which does not allow for the accurate discerning of acrophases. For example, Chrono and Per2 are not known to peak at the same time. The acrophase of Rora is doubtful.  

We appreciate that comment. We have studied the circadian expression conditioned by the photoperiod, and not the endogenous rhythm. For that purpose, the time course was collected during a controlled 12h/12h light/dark cycle and not in constant darkness. We understand the concern of the reviewer about the inclusion of more time points. However, a time course of 6 h interval has been widely use and accepted for the study of clock genes expression and rhythmicity, as broadly described in literature [1–5]. Viera et al, 2014 and Bonaconsa et al, 2014 discerned acrophases with time points obtained every 6 hours. Note that in this last work, Bonaconsa et al, 2014 studied acrophase and circadian rhythms in a 12h light/dark cycle. Therefore, our experimental design is justified and acrophase data are valid according to the provided references.

 References:

  1. Wang, X.; Wang, L.; Yu, Q.; Xu, Y.; Zhang, L.; Zhao, X.; Cao, X.; Li, Y.; Li, L. Alterations in the Expression of Per1 and Per2 Induced by Aβ31-35 in the Suprachiasmatic Nucleus, Hippocampus, and Heart of C57BL/6 Mouse. Brain Res 2016, 1642, 51–58, doi:10.1016/j.brainres.2016.03.026.
  2. Bonaconsa, M.; Malpeli, G.; Montaruli, A.; Carandente, F.; Grassi-Zucconi, G.; Bentivoglio, M. Differential Modulation of Clock Gene Expression in the Suprachiasmatic Nucleus, Liver and Heart of Aged Mice. Exp Gerontol 2014, 55, 70–79, doi:10.1016/j.exger.2014.03.011.
  3. Kolbe, I.; Leinweber, B.; Brandenburger, M.; Oster, H. Circadian Clock Network Desynchrony Promotes Weight Gain and Alters Glucose Homeostasis in Mice. Mol Metab 2019, 30, 140–151, doi:10.1016/j.molmet.2019.09.012.
  4. Murphy, B.A.; Blake, C.M.; Brown, J.A.; Martin, A.-M.; Forde, N.; Sweeney, L.M.; Evans, A.C.O. Evidence of a Molecular Clock in the Ovine Ovary and the Influence of Photoperiod. Theriogenology 2015, 84, 208–216, doi:10.1016/j.theriogenology.2015.03.008.
  5. Vieira, E.; Ruano, E. g; Figueroa, A.L.C.; Aranda, G.; Momblan, D.; Carmona, F.; Gomis, R.; Vidal, J.; Hanzu, F.A. Altered Clock Gene Expression in Obese Visceral Adipose Tissue Is Associated with Metabolic Syndrome. PLoS One 2014, 9, e111678, doi:10.1371/journal.pone.0111678.

 The conclusions and statements are often inconsistent, at times contradictory, about the effects of aging, melatonin, and NLRP3 deletion on several of the core clock genes. This is related to the poor design of the time course and poor data quality.

 The effects of aging, melatonin, and NLRP3 deletion on the core clock genes were assessed with two different analysis, which are complementary to each other: 1) Multifactorial-ANOVA analysis of the clock genes expression was used to reveal the main effect of time point, age, genotype and their interaction (Table 1, Figure 1, Figure 2, Supplementary Table S1); 2) Cosinor analysis was performed for the characterization of acrophase, amplitude, mesor and rhythm (Figure 3, Figure 4, Figure 5, Supplementary Table S2). The obtained data were cautiously interpreted and compared with the available publications. The design of the time course, and consequently our data quality, is supported by recent studies, as previously addressed.

  1. It is not clear how the expression pattern changes of these genes impact cardiovascular physiology.

 The aim of this manuscript was to study the impact of aging, NLPR3 inflammasome and melatonin therapy on the cardiac circadian system. Nevertheless, that is a very interesting point. To address this question, and characterize cardiac impairments, we took into consideration our data obtained in previous publications and performed light microscopy analysis (Figure 6). Results have been fully discussed in page 13 of the manuscript.

  1. There is no data presented that validates the NLRP3-/- model.

Indeed, we previously validated the absence of NLRP3 in the mutant mice used in our lab for various experimental models:

  • Rahim, I.; Djerdjouri, B.; Sayed, R.K.; Fernández-Ortiz, M.; Fernández-Gil, B.; Hidalgo-Gutiérrez, A.; López, L.C.; Escames, G.; Reiter, R.J.; Acuña-Castroviejo, D. Melatonin Administration to Wild-Type Mice and Nontreated NLRP3 Mutant Mice Share Similar Inhibition of the Inflammatory Response during Sepsis. J Pineal Res 2017, 63, doi:10.1111/jpi.12410.
  • Sayed RKA, Fernandez-Ortiz M, Diaz-Casado ME, Aranda-Martinez P, Fernandez-Martinez J, Guerra-Librero A, Escames G, Lopez LC, Alsaadawy RM , Acuna-Castroviejo D. "Lack of NLRP3 inflammasome activation reduces age-dependent sarcopenia and mitochondrial dysfunction, favoring the prophylactic effect of melatonin." J Gerontol A Biol Sci Med Sci;2019 74:1699-1708.
  • Sayed RKA, Fernandez-Ortiz M, Diaz-Casado ME, Rusanova I, Rahim I, Escames G, Lopez LC, Mokhtar DM , Acuña-Castroviejo D. "The Protective Effect of Melatonin Against Age-Associated Sarcopenia-Dependent Tubular Aggregates Formation, Lactate Depletion and Mitochondrial Changes." J Gerontol A Biol Sci Med Sci;2018 73:1330-1338.
  • Rahim I, K Sayed R, Fernández Ortiz M, Aranda-Martínez P, Guerra-Librero A, Fernández Martínez J, Rusanova I, Escamaes G, Djerdjouri B , Acuña Castroviejo D. "Melatonin alleviates sepsis-induced heart injury through activating the Nrf2 pathway and inhibiting the NLRP3 inflammasome." Naunyn Schmiedebergs Arch Pharmacol;2020 394:261-277.
  • Rahim I, K Sayed R, Fernández Ortiz M, Aranda-Martínez P, Guerra-Librero A, Fernández Martínez J, Rusanova I, Escamaes G, Djerdjouri B , Acuña Castroviejo D. "Melatonin alleviates sepsis-induced heart injury through activating the Nrf2 pathway and inhibiting the NLRP3 inflammasome." Naunyn Schmiedebergs Arch Pharmacol;2020 394:261-277.
  • Fernández Ortiz M, Sayed RKA, Fernández Martínez J, Cionfrini A, Aranda-Martínez P, Escamaes G, de Haro Muñoz T , Acuña Castroviejo D. "Melatonin/Nrf2/NLRP3 Connection in Mouse Heart Mitochondria during Aging." Antioxidants;2020 9:1-22.
  • Sayed RKA, Fernandez-Ortiz M, Fernandez-Martinez J, Aranda-Martinez P, Guerra-Librero A, Rodriguez-Santana C, de Haro T, Escames G, Acuña-Castroviejo D , Rusanova I. "The Impact of Melatonin and NLRP3 Inflammasome on the Expression of microRNAs in Aged Muscle." Antioxidants;2021 10:524.
  1. An old reference #26 (Belle et al, Cell Cycle, 2012) was used to support the statement that “The influence of NF-kB on the disruption of circadian rhythms during aging is well-established”. However, there have been several more recent papers, for example, Haspel et al, Nat Commun, 2014; Hong et al, G&D, 2018; and Shen et al, PLoS Genet, 2021.

Thank you. This issue was revised and corrected accordingly.

Reviewer 2 Report

The Authors in the present original study aims to assess the effect of aging, NLRP3 inflammasome and melatonin effect on the cardiac circadian system. Notably, melatonin improved the rhythmicity in cardiac muscle, emphasizing its potential clinical application in preventing/treating chronodisruption and sarcopenia aging-related.

            I think that the present manuscript has an interesting and actual topic and it could be a starting point to larger analysis for a better understanding on melatonin anti-aging/anti-inflammaging (heart) implications. In my opinion, this manuscript has a clear message, the rationale for the choice of the experimental model as well as the technical approaches used are appropriate. The obtained results are fully described and discussed.  

However, I have some mandatory comments:

In all the text check abbreviations (also in Abstract) and correct the few editing errors.

Introduction

- Page 2, line 59: please briefly define “chronodisruption”;

- Page 2, lines 62-66: briefly describe the cardiovascular dysfunctions NLRP3-related;

- Page 2, line 76: more than one reference is needed to justify the sentence.

Results

- Morphological analyses of heart tissue will be useful to better characterize cardiac alterations.

Discussion

- Insert a scheme that summarize the data obtain in the present study.

Materials and Methods

- Page 12, line 368: the sentence “irrespective of gender” is not clear;

- Page 12, line 371: specify the source of melatonin and give detail on melatonin food chow. 

Author Response

In all the text check abbreviations (also in Abstract) and correct the few editing errors.

Abbreviations have been checked and editing errors have been corrected.

Introduction

- Page 2, line 59: please briefly define “chronodisruption”;

This issue was revised and corrected accordingly.

- Page 2, lines 62-66: briefly describe the cardiovascular dysfunctions NLRP3-related;

This issue was revised and corrected accordingly.

- Page 2, line 76: more than one reference is needed to justify the sentence.

We highly appreciate that comment. That reference has been substituted by several other recent papers.

Results

- Morphological analyses of heart tissue will be useful to better characterize cardiac alterations.

We are grateful to the reviewer concerning this comment. A light microscopy analysis of cardiac muscle fibers has been included in our manuscript (Figure 6).

Discussion

- Insert a scheme that summarize the data obtain in the present study.

A scheme that summarizes the data obtained in the present study has been inserted in the Discussion section (Figure 7).

Materials and Methods

- Page 12, line 368: the sentence “irrespective of gender” is not clear;

We have corrected the sentence as follows:

Wild-type (WT) and NLRP3-/- male and female mice were classified into five experimental groups regardless of gender”.

- Page 12, line 371: specify the source of melatonin and give detail on melatonin food chow. 

 This issue was revised and corrected accordingly.

Round 2

Reviewer 1 Report

The authors did not directly address the concerns about the design and interpretation of the study. No other comments.